# Early Prediction of Adverse Pregnancy Outcome in Women with Systemic Lupus Erythematosus, Antiphospholipid Syndrome, or Non-Criteria Obstetric Antiphospholipid Syndrome

**DOI:** 10.3390/jcm11226822

**Published:** 2022-11-18

**Authors:** Núria Baños, Aleida Susana Castellanos, Giuseppe Barilaro, Francesc Figueras, Gema María Lledó, Marta Santana, Gerard Espinosa

**Affiliations:** 1BCNatal-Barcelona Center for Maternal-Fetal and Neonatal Medicine (Hospital Clínic and Hospital Sant Joan de Déu), Sabino de Arana, 1, 08028 Barcelona, Spain; 2Institut d’Investigacions Biomèdiques August Pi i Sunyer, Universitat de Barcelona, Carrer del Rosselló, 149, 08036 Barcelona, Spain; 3Department of Autoimmune Diseases, Hospital Clínic. Carrer de Villarroel, 08036 Barcelona, Spain

**Keywords:** systemic lupus erythematosus, antiphospholipid syndrome, non criteria obstetric antiphospholipid syndrome, pregnancy outcome, prediction, autoimmune disease

## Abstract

A prospectively study of pregnant women with systemic lupus erythematosus (SLE), antiphospholipid syndrome, or non-criteria obstetric antiphospholipid syndrome was conducted to describe the characteristics of women followed in a referral unit and to derive a predictive tool for adverse pregnancy outcome (APO). Demographic characteristics, treatments, SLE activity, and flares were recorded. Laboratory data included a complete blood cell count, protein-to-creatinine urinary ratio (Pr/Cr ratio), complement, anti dsDNA, anti-SSA/Ro, anti-SSB/La, and antiphospholipid antibodies status. A stepwise regression was used to identify baseline characteristics available before pregnancy and during the 1st trimester that were most predictive of APO and to create the predictive model. A total of 217 pregnancies were included. One or more APO occurred in 45 (20.7%) women. A baseline model including non-Caucasian ethnicity (OR 2.78; 95% CI [1.16–6.62]), smoking (OR 4.43; 95% CI [1.74–11.29]), pregestational hypertension (OR 16.13; 95% CI [4.06–64.02]), and pregestational corticosteroids treatment OR 2.98; 95% CI [1.30–6.87]) yielded an AUC of 0.78 (95% CI, [0.70–0.86]). Among first-trimester parameters, only Pr/Cr ratio improved the model fit, but the predictive performance was not significantly improved (AUC of 0.78 vs. 0.81; *p* = 0.16). Better biomarkers need to be developed to efficiently stratify pregnant women with the most common autoimmune diseases.

## 1. Introduction

Most autoimmune diseases are no longer an absolute contraindication for pregnancy. Advances in management of systemic lupus erythematosus (SLE) and antiphospholipid syndrome (APS) have provided a better quality of life of women at childbearing age and a consequent increase in the pregnancy rate among these women [1]. A group of special interest are those women who do not meet all clinic or laboratory diagnostic criteria for APS, which has been defined as non-criteria obstetric APS (NC-OAPS) [2]. Scarce data exist regarding their characteristics and prognostic factors despite representing a considerable number of women followed in high-risk obstetric units [3]. Despite management improvements, pregnancies in women with SLE, APS, or NC-OAPS still present a higher perinatal morbidity compared to the general population, especially related to placental insufficiency, such as preeclampsia (PE), intrauterine growth retardation (IUGR), or perinatal death [4,5,6]. The pathophysiology underlying obstetric morbidity remains elusive [7]. In fact, pregnancy is an exceptional situation in which multiple systems must successfully adapt to obtain optimal outcomes. Clinical or subclinical inflammation, activation of the complement, the presence of autoantibodies [8], hormonal dysfunction, and immune disturbances might contribute to alter these complex balance [9,10,11,12]. 

The reported risk factors for adverse pregnancy outcomes (PO) include presence of lupus anticoagulant (LAC), antihypertensive use, physician global assessment score greater than 1, and low platelet count [13]. Considering women with obstetric APS, LAC positivity seems to be the strongest predictor of APO [14,15], and triple aPL positivity has been associated to an especially high risk of pregnancy complications and thrombosis [16]. More recently, aberrant activation of the alternate complement pathway has also been associated to APO [9]. More specific mechanisms, such as incomplete downregulation of SLE-associated transcriptional networks, including type-l interferon (INF-I), have also been related to SLE complications during pregnancy [17]. Women with NC-OAPS have not been included in randomized clinical trials or in observational registries, and therefore, information about their risk of obstetric recurrence is scarce. 

Some studies have attempted to identify predictive variables, obtaining models with limited predictive capacity and diverse results [18,19,20]. We conducted a study based on a prospectively collected database of pregnant women with SLE, APS, or NC-OAPS in order to accurately describe the characteristics of women with the aforementioned autoimmune diseases and who were followed in a referral unit and to derive from these characteristics a predictive tool for adverse pregnancy outcome that can help clinicians in risk counselling and tailoring the pregnancy follow-up.

## 2. Materials and Methods

### 2.1. Study Design and Population

This is an observational cohort study based on a prospectively collected database of pregnant women with autoimmune diseases followed in a high-risk obstetric unit of a tertiary hospital between 2010 and 2019. Pregnant women diagnosed with SLE, APS, or NC-OAPS were included. Each pregnancy of the included patients was registered and analyzed as an independent episode. The diagnosis of SLE was established according to the criteria of the latest update of the American College of Rheumatology/EUropean League Against Rheumatology (ACR/EULAR) [21]. The diagnosis of APS was established according to Sydney updated criteria [22]. NC-OAPS was defined as aPL-related obstetric complications not fulfilling clinical and/or the laboratory classification criteria for APS [2]. 

The study protocol was approved by the local Ethics Committee (HCB/2020/0184), and all participants signed informed consent. All women were treated according to a specific protocol that included the schedule of visits, ultrasounds, and laboratory tests. Patients with aPL received treatment according to the latest recommendations [23,24,25].

### 2.2. Measures

Demographic characteristics, previous obstetric outcomes and episodes of thrombosis, pre-pregnancy arterial hypertension, mode of conception, presence of other autoimmune diseases, and treatments were recorded. Regarding SLE activity, SLE flare during pregnancy or puerperium was defined as onset or worsening of signs/symptoms of SLE in any organ or system attributed to the activity of the disease and that conditioned a change in the treatment [25]. In addition, Systemic Lupus Erythematosus Pregnancy Disease Activity Index (SLEPDAI) [26] and Lupus Low Disease Activity State (LLDAS) [27] were calculated as well as adjusted GAPSS (Global Anti-Phospholipid Syndrome Score) [28], which included presence of dyslipidemia, arterial hypertension, and status of LAC, aCL, and aB2GPI antibodies. Accumulated organ involvement of SLE (skin, articular, serosa, hematologic system, kidney, and nervous system) was also registered. 

Laboratory data included in each trimester of gestation a complete blood cell count, protein-to-creatinine urinary ratio (Pr/Cr ratio), C3, C4, and anti-dsDNA antibodies. Status of anti-SSA/Ro (anti-Ro) and anti-SSB/La (anti-La) antibodies were registered. Regarding aPL, anticardiolipin antibodies (aCL), anti-B2-glycoprotein I antibodies (anti-β2GPI), and LAC that tested positive at least twice on two or more consecutive occasions at least 12 weeks apart with titers >40 GPL/MPL (99th centile) and/or >40 AU, respectively, were recorded as criteria aPL positivity. Persistent aPL at low titers and those with medium-high aPL titers or LAC but not persistently positive (only one positive determination) were classified as non-criteria aPL positivity. 

Pregnancy and adverse pregnancy outcomes were recorded. APO were defined as follows: (a) fetal death > 10 weeks of gestation not explainable by chromosomal abnormalities, anatomical malformations, or congenital infections; (b) neonatal death before hospital discharge due to complications of prematurity and/or placental insufficiency; (c) preterm birth < 37 weeks due to placental insufficiency, gestational hypertension, or preeclampsia; (d) small for gestational age newborn (<percentile 5) in the absence of anatomical malformations or genetic alterations; and (e) preeclampsia during gestation or puerperium. 

### 2.3. Statistical Analysis

The data were collected and entered into a database for analysis using the STATA software (version 13.1; Texas, College Station). The distribution of data was evaluated using the Shapiro–Wilk normality test. Missing values in the first trimester protein-to-creatinine urinary ratio (*n* = 21) were imputed from maternal age and second-trimester values by expectation-maximization (EM) method assuming a normal distribution and checking for the randomness of missing values by Little’s statistic. The MVA-package of IBM SPSS 23.0 was used. The statistical significance of differences in continuous data was calculated using the Student’s *t*-test or the Mann–Whitney U-test for normally and not-normally distributed data, respectively. The categorical data was analyzed using Fisher’s exact test. Multivariable analysis was performed by selecting variables with *p* < 0.1 in the univariate analysis as potential predictors of APO. Logistic regression was applied to assess the OR and 95% CIs of APO for all potential predictors. A stepwise regression was then used (*p* < 0.05 for the forward and *p* < 0.10 for the backward steps) to identify baseline characteristics available before pregnancy that were most predictive of APO and to create the predictive model. All the models were runed by penalized maximum likelihood method, which is more robust in samples with rare events. From individual risks, receiver-operating characteristics (ROC) curves were plotted. The area under the receiver-operating characteristics curve (AUC) was used to assess the discrimination of the logistic model obtained. The same procedure was applied to identify and generate a model with variables available during the 1st trimester of pregnancy to assess if the addition of some variable to the pregestational model might improve the predictive ability. 

## 3. Results

A total of 188 women and 217 pregnancies were included in the study. The distribution of the diagnoses and incidence of APO according to the diagnosis is shown in Table 1. One or more APO occurred in 45 (20.7%) women. Baseline characteristics, medical and obstetric history, treatments, laboratory parameters, and SLE type are shown in Table 2. Regarding the presence of other autoimmune diseases other than SLE, APS, or NC-OAPS, there were 14 women who had another autoimmune disease. Specifically, there were 10 women with Sjogren syndrome: 9 of them in the SLE group and 1 in the NC-OAPS group. Three women also presented with systemic sclerosis: two in the SLE and one in the SLE+ aPL group. One woman in the SLE+ aPL group also suffered idiopathic juvenile arthritis. Therefore, most of the comorbidities were found in SLE group rather than in the APS or NC-OAPS patients. There were no differences in APO rate among women with more than one autoimmune disease. The distribution of APO are shown in Table 3. Obstetric follow-up and pregnancy outcomes are also shown in Table 3.

No differences were found regarding maternal age, body mass index, mode of conception and previous APO, or thrombosis episodes between study groups. Among the baseline epidemiological characteristics, non-Caucasian ethnicity (33.3% vs. 14%; *p* = 0.01), smoking (26.7% vs. 9.9%; *p* = 0.01), and pre-gestational hypertension (26.7% vs. 1.7%; *p* < 0.001) were significantly more prevalent in the APO group. Regarding laboratory parameters, hypocomplementemia (low C3 or C4) (37.8% vs. 22.1%; *p* = 0.04) and Pr/Cr ratio in the 1st trimester (93–74; *p* = 0.02) and 3rd trimester (216–100; *p* < 0.001) were significantly associated to APO. Regarding treatments, only the use of corticosteroids treatment at the beginning of pregnancy was significantly more prevalent in APO group (40% vs. 16.3%; *p* = 0.002). However, mean corticosteroids dose was similar between APO and non-APO group (10 vs. 7.4; *p* = 0.19). No association was found with the other treatments, namely LDA, LMWH, and hydroxychloroquine. In this cohort, almost 70% of women were treated with low dose of aspirin, 40% with low-molecular-weight heparin (any dose), and almost 60% with hydroxychloroquine. Regarding SLE activity, only SLEPDAI score was significantly higher in the APO group. Kidney involvement was the only SLE organ involvement significantly associated to APO. Surprisingly, in our cohort, aPL positivity (meeting or not APS criteria, persistent LAC, double or triple aPL positivity) of any type was not associated to APO. As expected, APO group had more caesarean sections and SLE flares. 

A baseline model including non-Caucasian ethnicity (OR 2.78; 95% CI [1.16–6.62]), smoking (OR 4.43; 95% CI [1.74–11.29]), pregestational hypertension (OR 16.13; 95% CI [4.06–64.02]), and pregestational corticosteroids treatment (OR 2.98; 95% CI [1.30–6.87]) yielded an AUC of 0.78 (95% CI, 0.70–0.86). Among the potential first-trimester laboratory predictors, only Pr/Cr ratio (log_10_-trasnformed) (OR 1.92; 95% CI [1.13–3.26]); *p* = 0.021) significantly improved the model fit (R^2^-Naelgerkerke 0.35 vs. 0.32; chi-square *p* < 0.001). However, the predictive performance was not significantly improved (AUC of 0.78 vs. 0.81; *p* = 0.16). Figure 1 shows the ROC curves. Calibration plots are shown as a Appendix A. The final multi-variable weighted models of predictors of adverse pregnancy outcomes are shown in Appendix A. Details regarding the predictive performance of APO for fixed false-positive rate cut-offs are shown in Appendix A. Interval validation step using bootstrapping to adjust for overfitting/optimism are also shown in Appendix A.

## 4. Discussion

In this study, we profiled the demographic, clinical, and laboratory characteristics of women representing the clinical spectrum of women with autoimmune diseases attended in a referral unit. The status of the AID before or at the beginning of the pregnancy has proven to be a key aspect in order to obtain optimal pregnancy outcomes [13]. In this study, a predictive model of adverse pregnancy outcomes was developed in order to counsel women with SLE, APS, and NC-OAPS. For a screen positive rate of 50%, a baseline model that can be used even before pregnancy presents a detection rate of 83% for adverse outcome. This could be translated into the clinical practice by the tool being able to label 50% of the women with the more common autoimmune diseases as low-risk for adverse outcome (8%), while the other 50% of women labeled as high-risk would concentrate 83% of the adverse outcomes. 

Regarding the demographic characteristics, only non-Caucasic ethnicity and smoking were significantly associated to adverse pregnancy outcomes. Similar were pregestational hypertension and corticosteroid treatment at the beginning of pregnancy, probably reflecting a non-stable disease before gestation. However, disease activity and its relation to poor outcomes was not reflected by LLDAS index or anti-dsDNA positivity. Only hypocomplementemia (C3 or C4) was associated to APO even though it was not finally included in the model. Similarly, Pr/Cr ratio at the 1st and 3rd trimesters was associated to APO, which might reflect some degree of kidney involvement, a marker of worse prognosis in women with SLE [29,30]. However, this parameter did not independently add predictive capacity to the baseline model. 

Another finding to remark is the absence of association between any APO and the positivity of any of antiphospholipid antibodies regardless of whether they met the diagnostic criteria for APS or not. These results differ from other studies, in which lupus anticoagulant positivity was the main predictor of adverse pregnancy outcome in women with aPL [14,19]. GAPPS score was also explored as a marker of APO in order to examine possible shared pathophysiological mechanisms between thrombosis and APO, but no association was found. In our cohort, only 9.7% of women with SLE (+/− APS or aPL) developed a SLE flare during pregnancy or puerperium, which is inferior to the rates reported in previous studies [31]. This could be explained by the fact that most of the women presented a stable disease at time of conception, with a median SLEPDAI score of 2. Most of them had a pre-conception consultation with counseling on the risks of pregnancy and treatment optimization. Furthermore, the follow-up was carried out in a specialized consultation, where a specialist in maternal fetal medicine and a specialist in autoimmune diseases jointly saw the patient and made agreed therapeutic decisions. Altogether, our predictive models have a goodness-of-fit far from optimal (explaining only about one-third of the uncertainty to present an adverse pregnancy outcome), which highlights the need to explore new biomarkers that would help individualize the management of these pregnant women, thus improving their perinatal results.

It is worth mentioning the inclusion of women with NC-OAPS, who have been routinely excluded from studies despite representing approximately one-quarter of the pregnant women with aPL positivity referred to high-risk units. The inclusion of this group in our model enhances the external validity. In our cohort, 21% of women with NC-OAPS presented an APO, which agrees with data from previous studies and confirms that women with such autoimmune diseases still present worse perinatal results. Regarding study limitations, sample size is small due to the relatively low prevalence of AID in pregnant women. When the number of events is small, model overfitting can be a problem. An overfitted model tends to demonstrate poor predictive accuracy when applied to new data, but this small-events problem may also undermine external validation. Although we applied frequentist shrinkage methods to alleviate overfitting (penalized methods), we acknowledge that the small number of events in our cohorts may still result in an underpowered validation. It also worth mentioning that each one of the pregnancies from the recruited women were included and analyzed as independent events. However, only twenty-nine women presented two episodes or pregnancies. No women with more than two pregnancies were included. 

However, advances in management of AID have provided a better quality of life with a consequent increase in the number of pregnancies among these patients, a tendency that will continue to grow in the upcoming years.

## 5. Conclusions

To conclude, until new and better APO biomarkers are developed, a model including ethnicity, smoking status, pre-gestational hypertension, and corticosteroids treatment at the beginning of pregnancy could efficiently stratify pregnant women with the most common autoimmune diseases according to their risk for subsequently presenting an adverse pregnancy outcome.

## Figures and Tables

**Figure 1 jcm-11-06822-f001:**
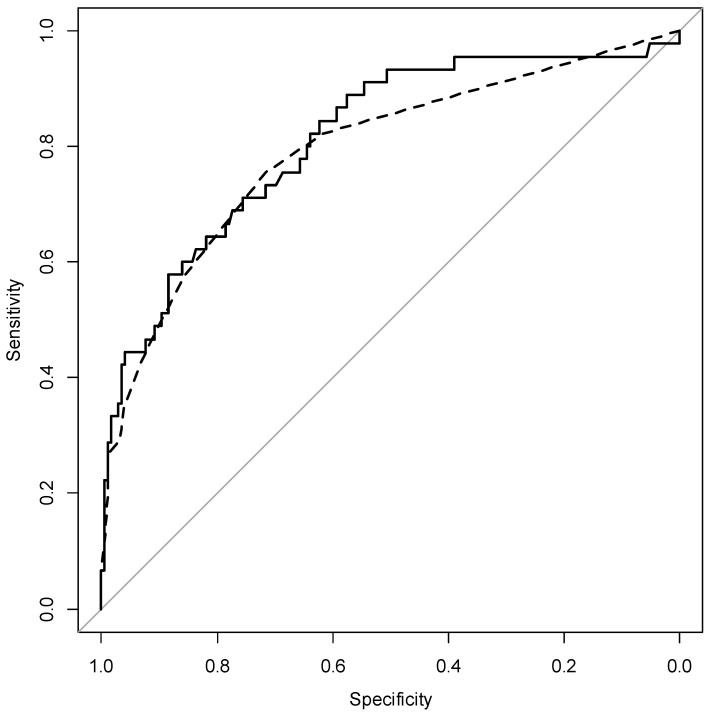
Comparison of ROC curves between the baseline model (dashed line) and the model including urinary protein (solid line).

**Table 1 jcm-11-06822-t001:** Distribution of pregnancies included and incidence of APO according to the diagnosis.

		Total *n* = 217	With APO *n* = 45	*p*
Diagnosis	APS	41	9 (22)	0.70
	NC-OAPS	42	9 (21.4)	
	SLE	100	18 (18)	
	SLE + aPL	26	6 (23.1)	
	SLE + APS	8	3 (37.5)	

Abbreviations: APO, adverse pregnancy outcomes; APS, Antiphospholipid syndrome; NC-OAPS, non-criteria obstetric antiphospholipid syndrome; SLE, Systemic Lupus Erithematosus; aPL, antiphospholipid antibodies. Data given as *n* (%).

**Table 2 jcm-11-06822-t002:** Characteristics of study population and their association with adverse pregnancy outcome.

	Total *n* = 217	With APO *n* = 45	Without APO *n* = 172	*p*
Maternal age	36.5 (33.5–39.5)	36.5 (33.5–39.5)	36.5 (33.5–39.5)	0.83
Body mass index	22.5 (20.5–24.7)	22.8 (20.7–25)	21.6 (19.2–23.7)	0.42
Non-Caucasian	39 (18)	15 (33.3)	24 (14)	0.01
Smoking	29 (13.4)	12 (26.7)	17 (9.9)	0.01
Assisted reproductive technique	39 (18)	12 (26.7)	27 (15.7)	0.08
Previous adverse pregnancy outcome	26 (12)	6 (13.3)	20 (11.6)	0.8
Previous thrombosis	18 (8.3)	4 (8.9)	14 (8.1)	0.8
Pre-gestational hypertension	15 (6.9)	12 (26.7)	3 (1.7)	<0.001
Other autoimmune diseases	14 (6.4)	3 (6.6)	11 (6.4)	0.94
LDA	150 (69.1)	38 (80)	114 (66.3)	0.1
LMWH	88 (40.6)	20 (44.4)	68 (39.5)	0.61
Corticosteroids beginning pregnancy	46 (21.2)	18 (40)	28 (16.3)	0.002
Corticosteroids dose (mg)	5 (5–10)	5 (3.3–10)	5 (5–10)	0.19
Hydroxychloroquine treatment	126 (58.1)	27 (60)	99 (57.6)	0.87
GAPSS	0 (0–4)	0 (0–4)	0 (0–4)	0.25
LLDAS	119 (54.8)	22 (48.9)	97 (56.4)	0.4
SLEPDAI	2 (0–4)	2 (0–4)	2 (0–2)	0.03
Anti-Ro/anti-La antibodies	57 (26.3)	11 (24.4)	46 (26.7)	0.85
Anti-dsDNA antibodies	78 (35.9)	19 (42.2)	59 (34.3)	0.38
Low levels of complement (C3 or C4)	55 (25.4)	17 (37.8)	39 (22.1)	0.04
Lupus anticoagulant Negative No criteria Criteria	176 (81.1) 10 (4.3) 31 (14.6)	32 (71.1) 4 (8.9) 9 (20)	144 (83.7) 6 (3.5) 22 (12.8)	0.11
aCL IgM Negative No criteria Criteria	180 (83) 8 (3.7) 29 (13.4)	36 (80) 2 (4.4) 7 (15.6)	144 (83.7) 6 (3.5) 22 (12.8)	0.71
aCL IgG Negative No criteria Criteria	166 (76.5) 22 (10.1) 29 (13.4)	36 (80) 6 (13) 3 (6.7)	130 (75.6) 16 (9.3) 26 (15.1)	0.25
Anti-β2GPI IgM Negative No criteria Criteria	186 (85.7) 15 (6.9) 16 (7.4)	37 (82.2) 4 (8.9) 4 (8.9)	149 (86.6) 11 (6.4) 12 (7)	0.67
Anti-β2GPI IgG Negative No criteria Criteria	196 (90.3) 9 (4.2) 12 (5.5)	43 (95.6) 1 (2.2) 1 (2.2)	153 (89) 8 (4.7) 11 (6.4)	0.57
Triple aPL positivity	13 (6)	4 (8.9)	9 (5.2)	0.36
SLE type Musculoskeletal Dermatologic Hematologic Renal Serosa Neurological	101 (46.5) 96 (44.2) 53 (24.4) 34 (15.7) 26 (12) 3 (1.4)	21 (46.7) 22 (48.9) 11 (24.4) 12 (26.7) 9 (20) 0 (0)	80 (46.5) 74 (43.0) 42 (24.4) 22 (12.8) 17 (9.9) 3 (1.7)	0.5 1 1 0.04 0.07 1
Protein/creatinine urinary ratio at 1T	77 (55–109)	93 (57–162)	74 (54–97)	0.02

Abbreviations: APO, Adverse pregnancy outcome; LDA, low-dose aspirin; LMWH, low-molecular-weight heparin; LLDAS, low-level disease activity state; GAPSS, Global Anti-Phospholipid Syndrome Score; LLDAS, Lupus Low Disease Activity State; SLEPDAI, Systemic Lupus Erythematosus Disease Activity Index; aCL, anticardiolipin antibodies; β2GPI, anti-B2-glycoprotein; aPL, antiphospholipid antibodies; SLE, Systemic Lupus Erithematosus; 1T, first trimester. Data given as *n* (%) or median (IQR).

**Table 3 jcm-11-06822-t003:** Obstetric follow-up and pregnancy outcomes.

	Total *n* = 217	With APO *n* = 45	Without APO *n* = 172	*p*
Pregnancy loss 10–24 weeks	4 (1.8)			-
Fetal loss > 24 weeks	4 (1.8)			-
Neonatal death	3 (1.4)			-
Preeclampsia	18 (8.3)			-
Placental abruption	1 (0.5)			-
Preterm birth	19 (8.8)			-
Small for gestational age	28 (13.3)			-
Onset of labor Spontaneous Induction Elective caesarean section	65 (31) 104 (49.5) 41 (19.5)	7 (16.3) 25 (58.1) 11 (25.6)	58 (34.7) 79 (47.3) 30 (18)	0.06
Mode of delivery Vaginal Non-elective caesarean section Elective caesarean section	122 (58.4) 46 (22) 41 (19.6)	17 (38.6) 16 (36.4) 11 (25)	105 (63.6) 30 (18,2) 30 (18.2)	0.01
Gestational age at delivery	39.3 (37.8–40)	37.1 (34–38)	39.6 (38.7–40.1)	<0.001
Birthweight	3090 (2700–3400)	2250 (1608–2653)	3155 (2980–3460)	<0.001
Thrombosis gestation or puerperium	3 (1.4)	1 (2.2)	2 (1.2)	0.5
SLE flare gestation or puerperium	13 (6)	6 (13.3)	7 (4.1)	0.03
Protein/creatinine urinary ratio at 2T	93 (63–130)	93 (72–168)	91 (62–122)	0.17
Protein/creatinine urinary ratio at 3T	114 (74–182)	216 (115–597)	100 (72–144)	<0.001

Abbreviations: APO, Adverse pregnancy outcome; SLE, Systemic Lupus Erythematosus; 1T, first trimester; 2T, second trimester; 3T, third trimester. Data given as *n* (%) or median (IQR).

## Data Availability

The original database is available to reviewers/editors and can be sent by the authors if required.

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
