# Peer review of "Early Prediction of Adverse Pregnancy Outcome in Women with Systemic Lupus Erythematosus, Antiphospholipid Syndrome, or Non-Criteria Obstetric Antiphospholipid Syndrome"

_jcm, 2022, doi:10.3390/jcm11226822_

Round 1

Reviewer 1 Report

The aim of this study was to develop a predictive model for the pregnancy outcome of women with autoimmune diseases, especially with evidence of anti-phospholipid autoantibodies. This is an important goal of clinical research so that these patients can be followed during their pregnancy in a risk-adapted manner. The novelty of this study is the inclusion of patients with NC-OAPS, for whom there are hardly any valid data on pregnancy outcome so far.

The manuscript is clearly structured and well written, however, there are a number of points to clarify:

Methods:

The description of the study population lacks information on possible other autoimmune diseases of the patients included with APS or NC-OAPS. This is particularly noticeable because SLE appears as a separate entity. Here it would be interesting to know whether the patients with APS or patients with anti-PL antibody detection without fulfilling the criteria for APS also have a specific rheumatological diagnosis (e.g. RA, Sjögren's, MCTD, were criteria used for diagnosis or for exclusion?). Or was the detection of anti-phospholipid antibodies alone sufficient to designate subjects as "pregnant women with autoimmune diseases"?

Statistics:

The question arises as to whether it would not be more appropriate to include persons with certain clinical and laboratory constellations and not the individual events in persons (in 13%, 2 pregnancies were included in the analysis) with their characteristics for the derivation of a predictive model.

In addition, it would have to be investigated whether the combination of certain anti-phospholipid autoantibodies (it is not uncommon for more than one autoantibody to be present) might be associated with APO.

Why did you not include the factor “autoimmune diseases” (APS, NC-OAPS (unmentioned: with or without another definite autoimmune diagnosis like rheumatoid arthritis, Sjögren´s syndrome, MCTD etc), SLE+APS) in the statistical model? There was a remarkable difference in the incidence of APO between diagnoses (see table 1).

The approach for selection of variables in multivariable analysis is not state of the art. Variable selection by a p-value cutoff (e.g. p<0.01) or classical backward/ forward selection algorithms (Whittingham et al (2006). Why do we still use stepwise modelling in ecology and behaviour?,  J Anim Ecol; 75(5):1182-9) is not state of the art and may lead to false biological models and conclusions. An alternative (modern) way of approaching the regression would be a LASSO regression (least absolute shrinkage and selection operator).

Results:

In the sentence: "a total of 217 pregnancies were included in the study." the number of patients should be mentioned.

Table 1: Looking at the line percentages, it reads like this to me:

18% of SLE patients (of the total cohort) have an Adverse Pregnancy Outcome (APO).

but:

77% of SLE patients who also have aPL antibodies and.

63% of SLE patients who also have antiphospholipid syndrome (APS), and

78% of patients with anti-PL antibody detection without meeting the criteria for aPS and apparently without SLE (other rheumatic diseases are not mentioned) had an Adverse Pregnancy Outcome.

Without statistical calculations, I would infer from Table 1:

Detection of aPL antibodies - with or without SLE - increases the risk of APO.

The percentages in Table 1 are not clear to me - in part I recognise column percentages, in part the numbers for this are not accurate.

Also in table 2 there are ambiguities in the percentages according to my understanding (aCL-IgG Criteria: once 6 patients are 13%, once 6.7%)).

Table 3: Shouldn't the contents that are in the first 7 rows be under the heading "APO" (and not under the heading "Total")?

And why was the structure "Total", "with APO" and "without APO" not used further?

Line 150: "No association was found with other treatments."

What other treatments were there among the patients in the study population (apart from glucocorticoids, hydroxychloroquine, aspirin, heparin)?

In the discussion, the following sentences appear contradictory to each other:

Line 203 “Another finding to remark is the absence of association between any APO and the positivity of any of antiphospholipid antibodies, regardless of whether they met the diagnostic criteria for APS or not.”

and

Line 224 “In our cohort, 21% of women with NC-OAPS presented an APO, which agrees with data from previous studies and confirms that women with such autoimmune diseases still present worse perinatal results.”

Clearer wording should be chosen here.

Author Response

Thank you. 

Reviewer 2 Report

The proposed manuscript : "Early-prediction of adverse pregnancy outcome in women with systemic lupus erythematosus, antiphospholipid syndrome or non-criteria obstetric antiphospholipid syndrome" is a relevant study which offers a predictive tool for pregnancy outcome in patients with autoimmune diseases. The proposed study can help clinicians in risk counselling and tailoring those pregnancies follow up. It profiled demographic, clinical and laboratory characteristics of women with clinical spectrum of women with autoimmune diseases.

The research question is valid and appropriate, methodology and interpretation of results are adequate. The results are analyzed and interpreted correctly. The method are appropriate and properly conducted, and the conclusions drawn is fully supported by the data. The manuscript is written well and easy to read. There  are no ethical concerns or other issues I believe the Editor should be aware.

Author Response

Thank you for your comments. 

Reviewer 3 Report

Manuscript ID jcm- 2017850 " Early-prediction of adverse pregnancy outcome in women with systemic lupus erythematosus, antiphospholipid syndrome or non-criteria obstetric antiphospholipid syndrome”

A study that evaluates the adverse pregnancy outcome of the pregnant women with systemic lupus erythematosus (SLE), antiphospholipid syndrome or non-criteria obstetric antiphospholipid syndrome. It is an important and an original study.

I have a few suggestions:

The p column should be added to Table 2.

The results of the non-APO group and the p column should also be added to Table 3.

The results of the logistic regression analysis should be given in the main text, not in supplementary file.

Author Response

Thank you. 

Round 2

Reviewer 1 Report

The comments have been sufficiently answered and the manuscript has been revised in terms of content in this respect.

However, one inconsistency remains:
According to Table 1, of the patients with NC-OAPS (n=42), a total of n = 33 should have had an adverse pregancy outcome (APO), but this would be 78.6% (row percent) of this patient group or 73.3% (column percent) of the patients with APS.

Presumably this is a confusion of the absolute numbers with the number of patients without APS?

Author Response

Dear reviewer, 

Thank for your comment and for  having detected the error. It was a transcription error. The number of patients with APO in NC-OAPS group was  9 (21.4%). We have corrected it in Table 1.